# Totally Excited about Moving Mobility and Exercise (TEAM M^e^): A Successful Multidisciplinary Exercise Quality Improvement Initiative for Pediatric and Young Adult Oncology and Stem Cell Transplant Inpatients

**DOI:** 10.3390/children9020186

**Published:** 2022-02-02

**Authors:** Angela M. Shaw, Rhonda Robert, Kimberly Kresta, Clark R. Andersen, Betsy Lewis, Brittni Maetzold, Raymund Valderrama, Ian Cion, Priti Tewari

**Affiliations:** 1Division of Pediatrics, The University of Texas MD Anderson Cancer Center, Houston, TX 77030, USA; amshaw12@gmail.com (A.M.S.); rrobert@mdanderson.org (R.R.); kmkresta@mdanderson.org (K.K.); bslewis@texaschildrens.org (B.L.); brittni.maetzold@gmail.com (B.M.); iancion@mac.com (I.C.); 2Department of Biostatistics, The University of Texas MD Anderson Cancer Center, Houston, TX 77030, USA; crandersen@mdanderson.org; 3Department of Physical Therapy, The University of Texas MD Anderson Cancer Center, Houston, TX 77030, USA; rvalderrama@mdanderson.org

**Keywords:** pediatric cancer, exercise, inpatient program

## Abstract

Background: Pediatric, adolescent and young adult (PAYA) patients are less active than their healthy counterparts, particularly during inpatient stays. Methods: We conducted a quality improvement initiative to increase activity levels in patients admitted to our pediatric oncology and cellular therapy unit using a Plan-Do-Study-Act (PDSA) model. An interdisciplinary team was assembled to develop an incentive-based inpatient exercise and activity program titled Totally Excited About Moving Mobility and Exercise (TEAM M^e^). As part of the program, patients were encouraged by their care team to remain active during their inpatient stay. As an additional incentive, patients earned stickers to display on TEAM M^e^ door boards along with tickets that could be exchanged for prizes. Activity was assessed by documentation of physical therapy participation, tests of physical function, and surveys of staff perceptions of patient activity levels, motivations, and barriers. Results: Compared to baseline, patient refusals to participate in physical therapy decreased significantly (24% vs. 2%) (*p* < 0.02), and staff perceptions of patient motivation to stay active increased from 40% to 70% in the post implementation period. There were no changes in physical function tests. Conclusions: An incentive-based exercise program for young oncology inpatients greatly improved patient activity levels, participation in physical therapy and influenced professional caregivers’ beliefs.

## 1. Introduction

Physical activity has been well-established as having short and long-term health benefits for almost all populations, while literature exists in adult oncology investigating the benefits of exercise, including decreased cancer-related fatigue, improved motor performance [1,2], and improved overall health-related quality of life [3] among both survivors of cancer and those receiving active therapy. In children, the data is much less conclusive. Fewer well-designed, appropriately powered trials exist, and most studies suffer from methodological weaknesses, small sample sizes and unclear or lack of intervention fidelity/exercise adherence. However, the available data suggests that physical activity may improve measures of physical function in children with cancer [4].

Unfortunately, children receiving therapy for cancer reach only 40% of the activity level compared to their healthy counterparts during their days at home. This number is further reduced during inpatient stays when children had less than one-fourth of the activity levels of their age-matched healthy counterparts [5].

The impetus to maintain fitness and regular exercise is a common roadblock, as generally few people have high self-motivation to be active and get exercise regularly, this can be even more challenging and significant an issue for the PAYA oncology patient population. Physicians, nursing, and rehabilitation staff can have extraordinary difficulty motivating this population to be active especially when they are ill.

To address this problem, the QI program described herein aimed to motivate and incentivize PAYA oncology and cellular therapy patients to increase physical activity levels with the ultimate goal to improve the health, fitness, symptom burden and quality of life for this high acuity patient population. We utilized a Plan-Do-Study-Act (PDSA) model to study the impact of this quality improvement program on the physical activity of children receiving cancer/stem cell therapy and on staff perceptions of patients’ physical activity. We report our findings in accordance with the Standards for Reporting Quality Improvement Excellence (SQUIRE) guidelines for reporting quality improvement studies [6].

The QI program, entitled “Totally Excited About Moving, Mobility and Exercise” (TEAM M^e^), was reviewed and approved by the institutional Quality Improvement Assessment Board (MDACC QIAB#731).

## 2. Materials and Methods

### 2.1. Planning and Context (‘Plan’)

An interdisciplinary team comprised of physicians, nurses, child life specialists, volunteer service staff members, a physical therapist, an occupational therapist, an artist in residence, and a clinical psychologist was assembled to elicit input on the problem of reduced physical activity in our pediatric inpatients and to develop an intervention. These team members provided extensive experience and background working and helping pediatric, adolescent and young adults undergoing treatment for cancer. Our inpatient pediatric unit is designed for the treatment of a wide age range of patients with cancer.

This team aimed to motivate patients to be more physically active. Utilizing backbones of consciousness-raising and contingency management within the Transtheoretical Model of Health Behavior Change, a program was developed to motivate and incentivize behavior change [7]. Contingency management interventions were chosen as they provide reinforcement for engaging in the desired behavior and have shown positive responses for exercise in select adult cohorts [8,9,10,11]. Reflecting the wide age range, this incentive-based physical activity program was developmentally graduated from early childhood through early adulthood. For example, patients were able to exchange their tickets for tiered prizes that were developmentally and age appropriate. The foundation of this intervention (TEAM M^e^) was based on the Orbit Model for behavioral treatment development, wherein this quality improvement program and data serve as a Phase IIa, ‘Proof-of-Concept’ [12]. Program roll-out included TEAM M^e^ education and information dissemination (written and videos) during patient tours, staff meetings, the annual multidisciplinary resource fair, and Family Advisory Council meetings.

### 2.2. The Intervention (‘Do’)

TEAM M^e^ is a self-reporting, semi-independent/asynchronous incentive-based exercise and activity program. Upon admission to the inpatient pediatric unit at MD Anderson, the admitting nurse reviewed information about the TEAM M^e^ program with the patient and family members. Education sheets, in English and Spanish, regarding TEAM M^e^ were included in the admission packet. Clinical nurse leaders, bedside nurses, and the medical team also discussed the program with patients and family members during daily bedside rounds.

TEAM M^e^ approved activities were all identified and defined by the interdisciplinary team members. Activity examples included walking, yoga practice, and completion of individualized physical/occupational therapy assignments. Patients notified an interdisciplinary care team member before their participation in a TEAM M^e^ activity, and upon completion of the activity, patients were awarded stickers for display on their door board along with prize tickets.

The door board consisted of a honeycomb pattern with hexagon outlines that were the same size and shape as the earned stickers. This concept allowed patients to not only earn their stickers, but also to create works of art on their door board as they displayed these stickers. TEAM M^e^ prizes underwent a similar level of attention, where prizes were made available for this variety of developmental/age groups (for example, Hot Wheels cars for toddlers, and adult coloring books or gift cards for older participants). Positive verbal reinforcement was provided by members of the multidisciplinary care team, aiming to complement the token economy.

### 2.3. Patient Population Selection, Primary Patient Data Collection, Staff Survey (“Study”)

The eligible population comprised pediatric, adolescent, and young adult oncology and cellular therapy inpatients between 2 and 26 years of age at MD Anderson who were admitted to the inpatient pediatric unit at MD Anderson during the pre-implementation and post-implementation periods. The pre-implementation period included May 2015 to September 2015 and the post-implementation period included October 2015 to July 2016. All patients admitted to the pediatric unit were encouraged to participate in the TEAM M^e^ program. Eligibility requirements for the quality improvement program were (1) the ability to ambulate and (2) having an active physical therapy consultation. While all admitted patients were able to participate in the unit program, patients were ineligible to participate if they were non-ambulatory, admitted to the pediatric intensive care unit, had recent surgery or required supplemental oxygen. As part of standard clinical practice, a physical therapist performed a 6-min walk test (6MWT) with admitted patients. Each patient could have multiple measures done during their admission; this includes measures of their participation/refusals. The distance a patient could walk on a flat surface in 6 min was measured (6MWD). At the completion of the 6MWT, the Modified Borg Dyspnea Scale rating was recorded. The scale ranges from 0 (nothing at all) to 10 (maximal) dyspnea on exertion with 3 being rated as moderate [13,14,15]. Patient refusals to participate in standard daily physical therapy sessions were also documented. These data were abstracted from physical therapy inpatient treatment encounter notes after session completion. No additional rehab testing measures outside of our unit standardized practice were conducted. All data were de-identified at the time of collection. Data collection occurred over three periods: four months prior to TEAM M^e^ rollout (‘pre’-implementation), 3 months following the institution of the TEAM M^e^ program (early post-implementation = Post1), and 6 months to 9 months after the initiation of TEAM M^e^ (late post-implementation = Post2). Any adverse outcomes associated with TEAM M^e^ were recorded through the institutional patient event reporting system and were assessed monthly by members of the multidisciplinary team.

A 16-item survey, entitled ‘TEAM M^e^ QI’ was created by the principal investigator and clinical psychologist, and used to assess staff perceptions of patients’ physical activity levels and physical activity barriers and motivators. This was a Pre/Post survey with which results prior to and following program implementation were compared. All inpatient facing staff and faculty in the Division of Pediatrics were surveyed. A Likert scale ranging from “strongly agree” to “strongly disagree” was used in eight questions; 6 items included ‘check all that apply’ to represent these perceptions, and the final question asked for additional comments. The exact questions are included in the supplement. Questions 5 through 10 were adapted from Vasudevan and colleagues’ [16] Barriers to Physical Activity Questionnaire for People with Mobility Impairments. The original five items were designed to be answered by adults who have mobility impairments; in our application, the items were adapted for use with professional caregivers (e.g., “You lack the motivation to be physically active” was modified to “My patients are motivated to stay physically active”). The surveys were administered at two time points: 1 month before the implementation of TEAM M^e^ and 2 months after implementation of TEAM M^e^. Surveys were conducted using survey software to staff members via e-mails that contained a survey link and were completed anonymously. Surveys could be conducted only once at each of the two time points by staff members. Survey results were viewable by the principal investigator and psychologist only.

### 2.4. Statistical Analysis

The measures from the 6 min walk test (distance in feet) and Borg (score out of 10) were modeled by mixed-effect analysis of variance with relation to time point (Pre, Post1, Post2), adjusting for age and sex as covariates, and blocking on patient to control for repeated measures within each time point (no patient was assessed at more than a single time point). Differences among time points were assessed by unadjusted contrasts. Approximate normality of model residuals was verified by normal quantile plots.

Incidence of refusal to participate was modeled by mixed-effect logistic regression with relation to age, sex, time point (Pre, Post1, Post2), and day following the initial date assessed, blocking on patient to control for repeated measures within each time point. More complex models including an interaction between time point and day following initial date assessed, as well as generalized additive mixed models, which utilized a penalized spline to accommodate potentially nonlinear association between incidence of refusal and time, were rejected due to higher Akaike Information Criteria.

A 95% level of statistical significance was assumed in all statistical testing. Statistical analyses were performed using R statistical software (R Core Team, 2020, version 3.6.3) [14]. Differences among time points in the analysis of variance models were estimated by contrasts using the “emmeans” package [17], with adjusted means weighted proportionally to marginal frequencies. Catseye plots [18] were produced using the “catseyes” package [19]. Data collected from staff perception surveys were summarized using standard descriptive statistics including frequencies, percentages, and means.

## 3. Results

### 3.1. Patient Demographics

In total, 43 patients participated over the three periods: Pre, Post1, and Post2. Patient demographics are summarized in Table 1.

### 3.2. Patient Physical Activity Measures and Participation in Physical Therapy

Table 2 shows results of the 6MWT along with patient modified Borg scores during the pre-implementation, early post-implementation and late post-implementation timepoints.

For the 6 min walk test, distance walked was not associated with time, as summarized in Figure 1. Distance walked was significantly associated with age (*p* = 0.013), but not sex (*p* = 0.91). With each additional year of age, distance walked increased by 48 feet. For the Modified Borg assessment, there was no significant evidence of association with time, as shown in Figure 2. There was no significant evidence of association with age (*p* = 0.74) nor with sex (*p* = 0.40).

Refusals to participate in physical therapy sessions were recorded during each period. Of note, some of these measures were repeated per subject per time point, but not between time points. There were 17, 17, and 9 unique patients at the Pre, Post1, and Post2 time points, respectively, totaling 43 patients with 117, 51, and 47 encounters at those time points.

In the pre-implementation period, patients refused to participate in 24% of physical therapy sessions, including walking. This number decreased to 15% in the early post-implementation period, but was not statistically significant (*p* = 0.41). However, in the late post-implementation period, one refusal occurred, which was 2% of all sessions. Figure 3 illustrates refusal probability over time. This was significantly reduced from the Pre-Implementation period, *p* = 0.013; furthermore, the odds of refusal at the Post2 time point were about 1/14 (0.07) the odds at the Pre time point.

Incidence of refusing rehabilitation therapy services was significantly associated with sex (*p* = 0.017), with males having about half the odds of refusal as females (Table 3). 

No adverse outcomes or injuries occurred to any of the participants at any time interval.

### 3.3. Staff Perception Survey Results

Eighty-five staff members (51% response rate) completed the survey 1 month before implementation, and 71 staff members (42% response rate) completed the follow-up post-implementation survey. In both surveys, the respondents represented broad clinical disciplines and included physicians, nurses, nursing assistants, physical therapists, social workers, pharmacists, and child life specialists. Respondents’ employment longevity with this institution ranged from less than 6 months to more than 10 years; modal tenure was 2–5 years.

Staff perception of patients’ motivation changed between the pre-and post-implementation periods (Figure 4). On the pre-implementation survey, 44% of respondents agreed or strongly agreed that their patients lacked confidence in their ability to be physically active. This number decreased to 35% in the post-implementation survey. The overall perception of peer-to-peer motivations also changed considerably. Before implementation, 59% of staff believed that their patients lacked peers to look up to who are physically active. This percentage decreased to 33% after implementation. In contrast to patient motivations, staff perception of barriers to physical activity and exercise did not change after the implementation of Team Me. Fatigue, lack of motivation and bad mood were perceived as psychological barriers by greater than 70% of staff in both periods.

Cumbersome intravenous poles and lack of exercise equipment on the floor were the most common organizational barriers reported both before and after implementation. Staff perception of physical barriers remained largely the same as well; more than 85% of staff in both surveys believed that nausea, pain, and weakness are the main physical barriers to patient exercise.

Before the implementation of TEAM M^e^, over 60% of staff believed low fitness levels to be a physical barrier to exercise, which decreased slightly to 50% after the initiation of the program.

## 4. Discussion

Exercise is possible during even the most intense chemotherapy for patients with pediatric cancer and even those undergoing hematopoietic stem cell transplantation [20,21,22,23,24]. Unfortunately, opportunities for and motivation to exercise are often overlooked during inpatient stays. We sought to improve mobility and activity levels in a challenging population to motivate pediatric and AYA patients admitted to our oncology and stem cell transplant unit through the development of a self-reporting, semi-independent, incentive-based exercise program. We found this program to greatly improve patient activity levels and participation in physical therapy.

The most noteworthy result from this program was that patient refusal to participate in physical therapy sessions significantly decreased from the Pre-Implementation phase to Post-Implementation Phase 2 (24% vs. 2%, *p* < 0.02). In addition, while comparing the pre and post staff survey response, the percentage of respondents who agreed or strongly agreed that patients were motivated to stay active during their inpatient stays more than doubled after implementation, with over two-thirds (70%) of respondents agreeing or strongly agreeing that their hospitalized patients are motivated to stay active.

While historically difficult to motivate, this statistically significant increased participation in inpatient rehabilitation services was interesting and promising for this patient population. Despite the increased participation in physical activity demonstrated, no significant differences were observed in 6MWD or Borg scores between the periods. The authors speculate this may be due to a couple of factors. As shown in Table 2, although the average 6MWT distance and Borg scores were not statistically different across groups, the ranges reveal lower minimum distances in the 6MWT in Post1 and Post2 compared with Pre-Implementation (120 ft., 150 ft. and 320 ft., respectively). This information coupled with the higher participation rates of 85% and 98% in Post1 and Post2 compared with 76% during Pre-Implementation suggests that even very debilitated patients were participating during the post-intervention periods.

These highly deconditioned patients on the inpatient unit who would have previously likely refused to participate in the 6MWT with their physical therapist now made attempts to participate in the 6MWT after the implementation of TEAM M^e^. Other factors may include the brevity of the timeframe of follow-up and the small sample size.

In a similar program at our institution, which used a control group, adult patients who participated in an incentive-based mobility program had a larger increase in 6MWT, 23.4 feet, by the time of discharge (*p* = 0.0447) and fewer hospital days (hazard ratio 1.65; *p* = 0.005) compared with patients who declined participation [25].

Similar to previously published studies, staff engagement in this quality improvement initiative is essential to the program’s success [26,27,28]. In our case, TEAM M^e^ was well received by clinical providers. The voluntary staff survey participation rate in the pre- and post-implementation periods, respectively, and the broad range of roles of the participants demonstrated a high level of engagement in the program. Staff investment was vital to inpatient participation and program improvement, as staff members were solely responsible for TEAM M^e^ implementation. We used a staff survey to assess our exercise culture and its malleability after TEAM M^e^ implementation. Staff reported that they believed nausea, pain, and weakness were the most common physical barriers and that lack of exercise equipment on the inpatient unit was a strong organizational barrier to patient participation in exercise activities. The barriers most reported by our staff were consistent with patient-reported barriers from the literature. Forty pediatric cancer patients receiving intensive treatment in Germany identified symptom burden, specifically, nausea, pain, and physical fatigue, as their primary exercise barrier. These same patients desired access to a sports therapist and sports activity areas [29]. This congruence of staff and patient perspectives indicates that ways to modify these physical and organizational barriers warrant further investigation.

There are some limitations to our data. One limitation is survey response bias. While the survey was sent out to all staff who interact directly with admitted patients, our response rates were 51 and 42%, respectively, allowing the possibility of sampling bias amongst the survey responders. Following implementation, responders were all readily aware of the TEAM M^e^ program, which was available to their patients and generally asked to encourage patients to participate. Furthermore, the lack of a comparator arm makes it difficult to conclude the active components of the intervention. Additionally, the 6MWT, a frequently used outcome measure in both pediatric and adult exercise studies, has pediatric norms that vary by patient age and sex, however, very few studies transform raw 6MWD into standardized numbers, such as Z scores, to avoid bias based on these age and sex differences [14]. Other limitations include the small sample size and generalizability to other pediatric oncology inpatient units. Our population was skewed to adolescents and young adults with cancer, collectively classified as AYAs, a group considered to be an underserved population because neither pediatric nor medical oncologists have previously focused on their specific care needs [30]. This lack of focus on AYAs has historically contributed to lags in survival gains and HRQL outcomes for this population leading to a surge of AYA programs in pediatric oncology centers. Our institution, for example, has a highly organized and well-supported Adolescent and Young Adult (AYA) program for patients primarily with pediatric type tumors/cancers. A large majority of these patients are treated by pediatric oncologists and are treated on our pediatric inpatient unit. Our pediatric therapists, physicians, nurses, psychologists, and even child life specialists are specifically trained to work with children and AYAs to provide optimal support to this underserved population.

The aim of this initiative was to motivate participants to increase their physical activity with the long term goal to improve the health and fitness of this high acuity patient population. Following the success of this quality improvement initiative, the Team M^e^ investigators have launched ORBIT Phase IIb in inpatient children, adolescents and young adults undergoing stem cell transplant (‘Act’). This prospective, Phase IIb pilot study uses a comparator arm to clarify the effects of the TEAM M^e^ intervention on physical and psychosocial outcomes in pediatric stem cell recipients and is currently enrolling participants.

Overall, the TEAM M^e^ quality improvement program demonstrated that an incentive-based, pediatric, adolescent and young adult focused multi-disciplinary program to improve patient mobility and participation in physical therapy is effective and well-received. Future studies will determine if this intervention can lead to improved health-related outcomes in this population.

## Figures and Tables

**Figure 1 children-09-00186-f001:**
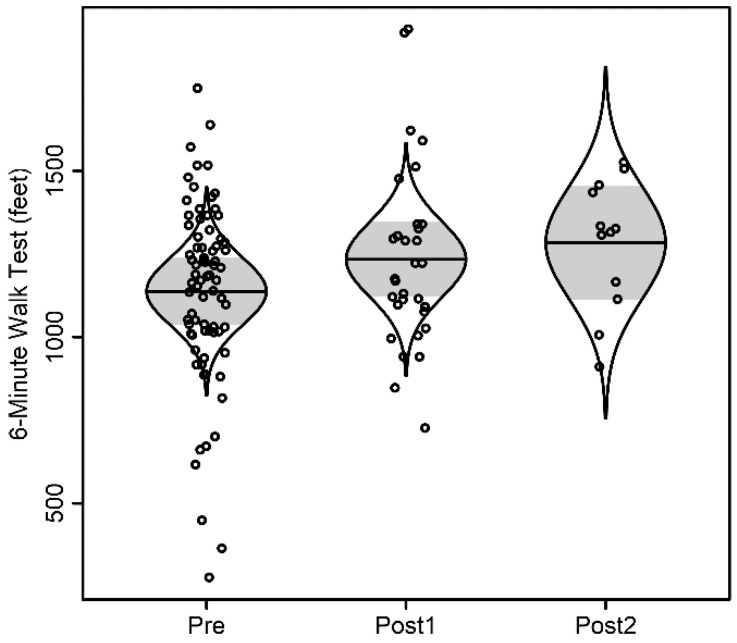
6 Minute Walk Test scatterplot. The model-adjusted scatterplot is randomly jittered horizontally for clarity and overlaid by catseye plots illustrating the normal distribution of the model-adjusted means with shaded +/− standard errors.

**Figure 2 children-09-00186-f002:**
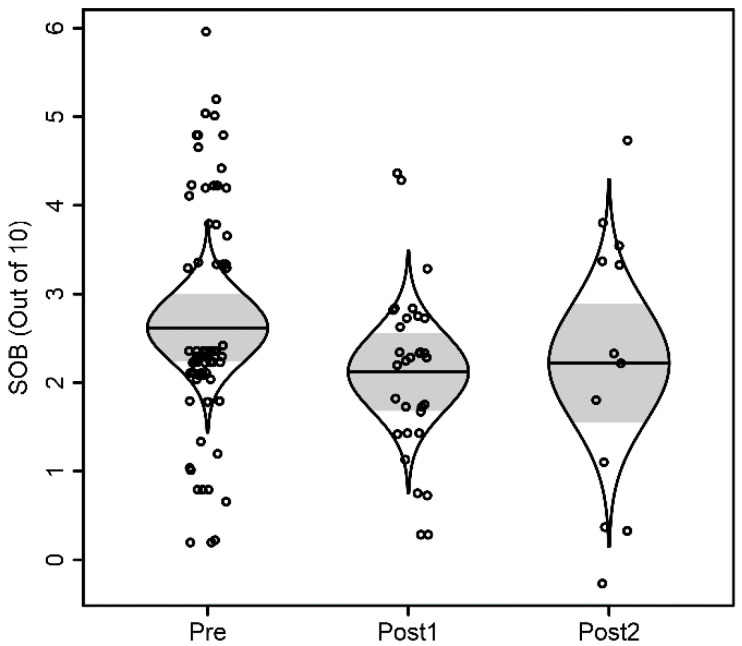
Modified Borg Score scatterplot. The model-adjusted scatterplot is randomly jittered horizontally for clarity and overlaid by catseye plots illustrating the normal distribution of the model-adjusted means with shaded +/− standard errors.

**Figure 3 children-09-00186-f003:**
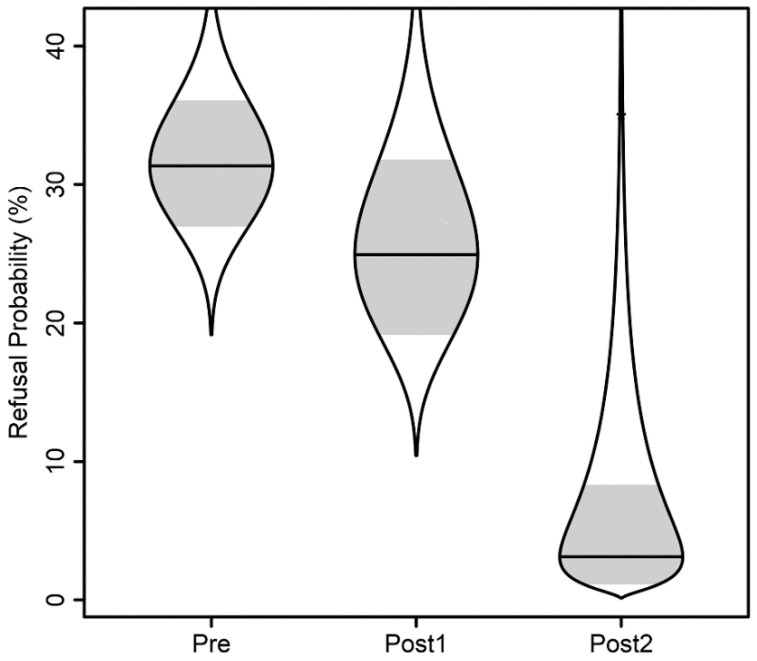
Refusal probability over time: logistic regression adjusted probability of refusal over time, controlling for age, sex, and day following the initial date assessed. Catseye plots illustrate the normal distributions of the model-adjusted means, with shaded +/− standard error intervals, transformed to the probability scale.

**Figure 4 children-09-00186-f004:**
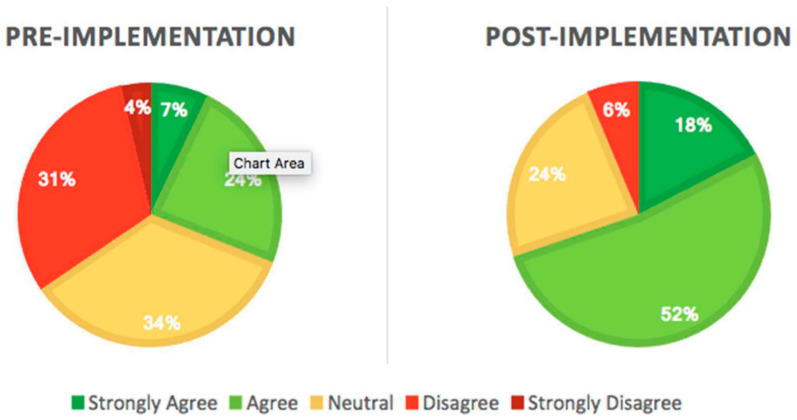
Staff survey result, pre/post staff survey results, staff responses to the statement “my patients are motivated to stay physically active”.

**Table 1 children-09-00186-t001:** Participant demographics.

	Total	Pre-Implementation (Pre)	Early Post Implementation (Post1)	Late Post Implementation (Post2)
Number of Patient	43	17	17	9
Number of Measures	215	117	51	47
Mean Age (SD)	18.2 (4.1)	18.9 (4.5)	17.4 (4.2)	18.6 (3.1)
Male (%)	24 (55.8)	9 (52.9)	8 (47.1)	7 (77.8)

**Table 2 children-09-00186-t002:** TEAM M^e^ QI Project patient physical activity levels and Six-Minute Walk Test results.

Measure	Pre-Implementation	Early Post Implementation (Post1)	Late Post Implementation (Post2)
Mean 6MWT Distance in	1238.6 (320–2100)	1323.3 (120–2300)	1185.3 (150–1860)
Feet (Range)			
Mean Modified Borg	2.17 (1–9)	2.13 (0–6)	2.12 (0–5)
Score (Range)			

**Table 3 children-09-00186-t003:** Refusal of Rehab Participation Analysis.

	Odds Ratio	CI95 Min	CI95 Max	*p*-Value
(Intercept)	0.57	0.12	2.61	0.46
Age	1.01	0.93	1.10	0.82
Sex (Male/Female)	0.42	0.21	0.85	**0.017**
Time (Post1/Pre)	0.73	0.34	1.55	0.41
Time (Post2/Pre)	0.07	0.01	0.57	**0.013**
Day	1.00	1.00	1.01	0.15

Bolded *p*-Values demonstrate statistically significant values from this analysis.

## Data Availability

Data sharing is not applicable to this manuscript.

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
