# Peer review of "Totally Excited about Moving Mobility and Exercise (TEAM Me): A Successful Multidisciplinary Exercise Quality Improvement Initiative for Pediatric and Young Adult Oncology and Stem Cell Transplant Inpatients"

_children, 2022, doi:10.3390/children9020186_

Round 1

Reviewer 1 Report

Dear authors,

thank you very much for this manuscript. I have the following questions/ comments:

  1. abstract: "no significant changes in 6 min walking test", but you conclude "greatly improved patient activity levels" - could you please explain?
  2. materials and methods: the study was conducted in 2015/ 2016 - why such a delay in publishing? are the findings still valid today?
  3. materials and methods: in principle do all patients have physiotherapy at your inpatient ward?
  4. results: 43 patients seem to be a low number for a big inpatient unit - how many were eligible? only 9 in the post 2 - bias possible?
  5. figure 3: percentage or number of patients? So more patients refused than participated in pre-implantation, but would have been eligible? That would be a high number which influences your results.
  6. results: as it was an incentive-based exercise program - any data on ongoing exercise as outpatient or at home? Long lasting effects?
  7. discussion: besides your phase IIb in SCT any other exercise projects implemented in the meantime as this study was completed 5 yrs ago?

Author Response

Dear Reviewer,

Thank you for your review and thoughtful questions/critique.  Attached are our responses.

Please See Attachment: 'Reviewer 1 response.1.2022'

Reviewer 2 Report

The authors have presented a before and after study examining some of the impacts of implementing a rewards-based intervention to promote mobility and activity in primarily young adults admitted to hospital with oncologic diagnoses or the need for immunotherapy.

I applaud the authors on the implementation of such a program and in their attempts to promote mobility and activity. My most significant issue with this study and publication in a journal called “Children” that is targeted to pediatrician readership is that there were not many children in the study; it is a study of adolescents and young adults and their motivators.

Major points:

  1. The study would benefit from further description of the origins and concept for the program. To me, an incentive-based program consisting of stickers and trading for prizes does not make intuitive sense for a population with a mean age outside the pediatric realm.
  2. The introduction is 2 paragraphs, convincing the reader that exercise is important and underperformed in the PAYA oncology population. I would tend to shift the focus, as it is not really necessary to convince most readers of the importance of exercise, or that it is difficult to get many oncology inpatients to do this – the whole intro as it stands might be summarized in 4-5 sentences. Then talk more about the potential approaches to increasing activity where their program fits within existing literature.
  3. The terminology throughout the paper would benefit from some clarity and consistency, which speaks to the need to more clearly define this paper within quality improvement literature. The clinicians implemented TEAM Me as a Quality Improvement program. The acceptability and effectiveness were then studied as a research study. I would avoid the use of the word “project”, which is interchangeably used to refer to the intervention being implemented and to the research/study component, and I would avoid calling the research component a quality improvement study; rather, I’d suggest that the team refer to the entire process in terms of a Plan-Do-Study-Act cycle (if that is what they did, and it seems like they did as there was clearly pre-implementation planning, implementation, and studying of the impacts).
  4. Without being firmly rooted in clearly-described quality improvement methods, I question the involvement of the participants and use of their data without the provision of informed consent. Were the walk tests standard for everyone or just those eligible for the study? Were there other procedures that were done with the participants that were specific to those who fit criteria for the study? This needs to be more thoroughly described.
  5. The statement of study objective needs to be changed. The program goal was to develop and implement and incentive-based PA program. The objective of the study was to assess some of the impacts of implementing the QI program. This is the study component of a PDSA cycle. If the objective is to report on the development and implementation of the program, and some of the impacts of implementation, then it needs stated as such and the results section needs to include more information about the process of development and implementation. Generally, the objective needs stated at the end of the intro, and the results need to address the objective.
  6. Unfortunately, I would say that the 18-25 year old population would be far better reported in the adult literature or perhaps in a specific pediatric oncology periodical where the readership would be oncologists rather than more general pediatricians, who may struggle to find relevance in this report. While I understand that the pediatric oncology population now includes young adults who are being managed for mainly blood cell line tumors, they are not pediatric patients. The developmental age and stage, motivators, and decision-making will all be different in a population that is predominantly young adults compared with a more usual distribution on a pediatric oncology floor that tends to be in the 4-16 age range. The mean age was 18.3 (outside the pediatric age group). The authors have not reported the standard deviation, only the age range of the participants, so it is not possible to completely understand whether the ages are mainly 16-20 with a few outliers at 10 and 25, or if the distribution is wide, encompassing more of the adolescents.
  7. Survey results: The staff survey reporting focuses on staff perception of the amount of activity being done, but the authors reported the actual percent change in activity level, so these results are not really important. Therefore, I would eliminate Figure 4A and any references to staff perceptions related to amount of activity. Far more relevant to report would be questions related to: staff perceptions on motivation (Figure 4b), related to staff opinions about the program, staff perceptions of barriers and enablers to mobility, whether the program changed the healthcare providers’ opinions about exercise in the population, and about whether staff members think that their population is capable of activity. I do not seem to have access to the supplement, so I cannot see the exact questionnaire to be able to tell if the authors are able to report these elements. The authors sometimes report agree/disagree, and sometimes specify what is “strongly agree” vs just “agree”. The authors had an open ended question for other thoughts at the end of the survey. How did the authors analyse these results? Are they planning to report them?
  8. Results section generally: All results from tables are repeated within text, and most of the tables and figures are repetitious. The authors should choose to report in either text or table, not both. Tables 3 and 4 are unnecessary, or they could be put together into 1 table and eliminate figures 1 and 2. Figure 3 is unnecessary.
  9. The number of participants is small, but the authors are reporting standard errors that would indicate much larger populations, or that the unit of measurement is number of PT sessions or invitations. However, they have not reported numbers for this. I’d suggest turning table 5 into a table more generally about refusal to participate. Then add a column for % refused. For each row (total, male sex, pre-intervention, post-intervention 1, post-intervention 2, etc) indicate the denominator used to calculate this.
  10. Lines 331-333 – this is important to describing the study design and rationale. This needs expanded and moved to either the introduction or to a section describing development of and rationale for the intervention.
  11. The conclusions mention feasibility of the program, but the manuscript did not describe any feasibility outcome measures that were assessed, there were none reported, and there was no mention of the resources required to implement this program and so the reader cannot reasonably infer feasibility. In addition, the lack of toddlers and children in the study would lead some to question the feasibility of the intervention.

Minor points:

  1. Line 68 – The study needs a description of the clinical unit involved. This would perhaps help to describe and explain the study demographic, which is different from that of any pediatric oncology units in which I have worked.
  2. Line 102 – how does the team intend to assess the effectiveness of the intervention on patients with altered forms of mobility?
  3. Line 102 – Were measures taken to ensure PT consultations were initiated? Is there a reason related to consultation that younger children would be less likely to be studied? The authors should help the reader understand why there are no children age 2-10 in the study.
  4. Line 123: There is a description that all “inpatient facing staff”. What does this mean>
  5. Line 123: Were all faculty in the division of pediatrics surveyed? All general pediatricians, and sub-specialists? Why? Do they care for inpatients in oncology?
  6. The authors have left a short paragraph at the end of the results that includes instructions on how to formulate the discussion.
  7. Line 224 – to how many staff members was the survey sent? I’d suggest looking at EQUATOR reporting guidelines for the survey to ensure you include all the necessary pieces of information.
  8. Discussion lines 280 and 284 – The authors should avoid hyperbole. The numbers are too small to indicate a true effect from the intervention, and if this was my study I would be very concerned that the mere act of telling patients about physical activity on admission and reinforcing it regularly, and requiring them to check in when they go for a walk would account for the change, rather than any particular incentives. This should be examined in future study that the authors are planning, and should be discussed.
  9. Line 319 – There are multiple types of survey bias. To which type is the author referring? Please be more specific here.
  10. Line 319-320 – The authors talk about survey respondents being unblinded. It seems that they are stating that it is a type of survey bias, though this would be incorrect. Also, blinding is not really relevant in a QI project like this.
  11. Line 328 – Agree that small sample is a limitation. This needs expanded. Why was the sample so small?

Author Response

Dear Reviewer,

Thank you for your review and the extensive time you spent critically evaluating this manuscript.  We appreciate your feedback and suggestions.

Please see attachment: 'Review 2 Response 1.2022'

Round 2

Reviewer 1 Report

no further comments or suggestions

Author Response

Dear Reviewer,

Thank you for taking time to review our manuscript and your thoughtful feedback and questions.

Reviewer 2 Report

I appreciate the opportunity to review this again.

The authors have made some important and necessary changes to the manuscript, which now reads better. In general, I believe that the authors have done good work, and have researched it appropriately, and are in need of some assistance with manuscript preparation to ensure that the reporting does the work justice.

I appreciate the authors’ justification of the age range, and am convinced that this underserved population should be studied and that it is a bit of an orphan population that needs “a place to go”. I would still encourage the authors to describe this in the discussion. I suspect that many readers will have the same response as I, particularly pediatricians who will wonder why this matters to them, and the authors’ description of the importance would be a welcome addition to the discussion.

The primary issue with this manuscript remains a need to properly report a quality assurance initiative and its assessment. The authors have added the term “PDSA cycle” throughout the manuscript, but are not using it correctly. They imply that each participant participates in a PDSA cycle. Unless the team continually assessed the intervention and provided small changes after each participant and the interpretation of their data, this is incorrect.

Rather than think of this manuscript as reporting a study on its own, the authors should think of it as reporting the entire body of their work. First, they designed and developed a program. Then they implemented it. Then they studied its impacts, and now their future work will be to make tweaks based on what they learned and to assess it in different contexts. The headings and subheadings in the manuscript need to reflect the different phases of this entire body of work, and not just the study part. If, however, you have reported the development and implementation elsewhere, then send the reader to that reference and don’t report it at all in this manuscript.

For a description of a quality assurance process that uses PDSA cycles, the authors need sections for:

Planning: They have planned an intervention (the entire description of the reasoning, the theoretical modeling, the use of a multi-disciplinary team, etc).

Do: Then they did the intervention – here they need to describe what they did, how they did it, for whom.

Study: Then they studied it – here they talk about the participants whose data was eligible for inclusion in the study, how they studied them, the results. They should also include the survey information here, as these are all about studying the impact and results and perceptions of their QI intervention.

Act: This will be about how they plan to change the intervention based on feedback and learnings from the study phase.

The authors should only state that they used PDSA cycles if they did, in fact, use them. Otherwise, they have used methods of assessing the impact of their Quality Improvement initiative.

In any case, the authors need to use a standard reporting guideline from the EQUATOR network. The SQUIRE checklist would assist them in more thorough and standardized reporting of a quality assurance initiative: https://www.equator-network.org/wp-content/uploads/2012/12/SQUIRE-2.0-checklist.pdf

The document would benefit from being quite a bit shorter with an attempt to remove extra words wherever possible. As a small example, lines 214-219 could read: We included 43 participants across 3 periods in the study component of TEAM Me who were primarily adolescents and young adults (mean [SD] age 18.2 [4.1] years, range 10-25 years) and 58.8% male (though I would have tended to keep this information tabular, removed the section heading, and merely stated: Patient demographic information for the 43 participants is summarized, Table 1.)

Suggest using a program like Grammarly that will eliminate word redundancies, of which there are quite a few (for example, “a sizeable amount”, line 37; “this specific population”, line 66).

Introduction:

Is the following really a reference for this work? Cultural Identity in Minoan Crete: Social Dynamics in the Neopalatial Period. Seems unrelated.

Methods:

I would avoid justifying care-based or research-based decisions on the concept of “we did a lot of work” or “we put a lot of thought into it”. Many teams do a lot of work or a lot of thinking that is still poor quality, and I would remove any reference to this concept from both the paper and responses to reviewers. What is important is the skill set and experience of their study team, and it is important information for the reader that the intervention was designed using a multi-disciplinary, inter-professional team of experts.

Section 2.3 – Although the survey was a short component to this broader program, it would be important to ensure that reporting of methods are consistent with standards. Look into the EQUATOR guidelines on survey reporting.

Lines 176-183 – this is not necessary, and its removal would shorten the manuscript appropriately. The authors have guided the reader to the appropriate reference and have included the survey as a supplemental file, so this can be removed.

I do not understand the new line at the end of the statistical methods. The sentence prior indicates how the survey results are reported (descriptively). If the authors mean to indicate that the survey results represent staff perceptions at the end of all three eras, this should be stated in the survey methods.

Results:

The authors removed Table 2, which I would have tended to keep, and instead removed the extra wording. Your reader is more likely to look at a table than to read text.

At line 240 the authors have added important information. The description of using multiple measures from each participant needs to go into methods. I do wonder whether this data, along with the demographic info, would be better in a table – one column for each era, a row for number of participants, row for number of measures, row for mean (SD) age, row for sex.

Table numbers need updated for changes to the manuscript.

Figure at lines 292 – 294 has only 1 description, but it looks like there are 2 pre-post comparisons.

Lines 300-310 seem to replicate what is in the figure and should be removed. Comment on the significance and importance can go into the discussion.

Author Response

Dear Reviewer,

Thank you for your very thoughtful suggestions.  We have incorporated these.  Your review and time spent on this manuscript/process has strengthened the manuscript significantly while really allowing us to share our work in this field.

Attached is our response along with a new version of our manuscript.  This includes a tracked copy and Clean copy of the manuscript.
